# Long-range optical coupling with epsilon-near-zero materials

Danqing Wang [1,2,3,4] ✉, Zheyu Lu[4,5,6], Sorren Warkander[2], Niharika Gupta [7], Qingjun Wang[2], Penghong Ci[2,8], Ruihan Guo [2,4], Jiachen Li [2], Ali Javey [4,7], Jie Yao [2,4], Feng Wang [4,5] & Junqiao Wu [2,4] ✉

Long-range resonant quantum tunneling of electrons happens across potential barriers when the wavefunction interferes constructively outside the barrier. Here we demonstrate an analogy in optical systems based on epsilon-near-zero materials, achieving phase-modulated, long-range optical interactions between transparent semiconducting oxide layers beyond the evanescent photonic coupling. Distinct from weak thin-film interference, intense electromagnetic fields confined within the epsilon-near-zero thin films show anti-correlated intensity oscillations as a function of interlayer separation up to hundreds of microns. The oscillatory, anti-correlated electromagnetic field intensities were probed by second harmonic generation from wedged indium tin oxide multilayers. Such a system that hosts subwavelength mode footprint and simultaneously long-range radiative coupling offers prospects for long-distance optical communication, large-scale photonic circuits, and hybrid quantum photonic systems.

Wavefunctions of Cooper pairs can tunnel through a non-superconducting barrier over distances longer than the coherence length[1,2]. Such a proximity effect results in an oscillatory dependence of superconducting transition temperature on the interlayer separation[3–5]. Besides a single potential barrier, electronic resonant tunneling with unity transmission happens in multilayer quantum barriers when constructive interference occurs at proper energy levels and geometric configurations with a characteristic tunneling length at the nanometer scale[6,7].

In optical systems, short-distance evanescent wave coupling happens at subwavelength scales, such as confined electromagnetic fields in plasmonic nanogap structures and mode crosstalk between dielectric waveguides[8–11]. On the other hand, long-range optical interactions can be enabled by radiative coupling to the far field in contrast to the evanescent wave decay of electromagnetic fields as a function of

distance[12,13]. Plasmonic nanoparticle superlattices support multiscale diffractive coupling among micron-spaced patches[14,15], and multilayer nanoparticle lattices sustain far-field interactions[16]. Furthermore, non-local interactions in nanostructured photonic moiré lattices introduce optical solitons and flat bands[17–20]. Such long-range optical coupling is desired for applications in long-distance optical communications, scalable integrated photonics, and large-scale quantum coherence[21–24]. The reported structured systems, however, rely on thousands of nanoscale building blocks or more for collective optical behavior, and the nanofabrication involves multi-step lithography approaches.

Epsilon-near-zero (ENZ) or near-zero-index materials emerge as a new class of materials that show exotic optical properties at the interface with dielectrics[25–29]. Nearly-zero phase delay within the ENZ matrix results in distinct optical responses, such as geometry-invariant light tunneling[30–35], optical cloaking[36–38], topology photonics[39,40], and

[1]College of Future Information Technology, Shanghai Engineering Research Centre of Ultra-precision Optical Manufacturing, Fudan University, Shanghai, China. [2]Department of Materials Science and Engineering, University of California, Berkeley, Berkeley, CA, USA. [3]Miller Institute for Basic Research in Science, University of California, Berkeley, Berkeley, CA, USA. [4]Materials Sciences Division, Lawrence Berkeley National Laboratory, Berkeley, CA, USA. [5]Department of Physics, University of California, Berkeley, Berkeley, CA, USA. [6]Graduate Group in Applied Science and Technology, University of California, Berkeley, Berkeley, CA, USA. [7]Department of Electrical Engineering, University of California, Berkeley, Berkeley, CA, USA. [8]State Key Laboratory of Semiconductor Physics and Chip Technologies, Institute of Semiconductors, Chinese Academy of Sciences, Beijing, China.
✉e-mail: danqingwang@fudan.edu.cn; wuj@berkeley.edu

directional thermal emission[41–44]. In addition, as driven by the continuity of the electric displacement field **D** at the interface, electromagnetic waves can be highly confined and strongly enhanced within ENZ thin films under transverse magnetic (TM) polarization at oblique incidence[45–47]. Intense near fields lead to enhanced second harmonic generation (SHG) in transparent semiconducting oxides, such as indium tin oxide (ITO) thin films[48–51], which support Berreman absorption peak[52] at ENZ wavelengths in the near infrared. A metal/insulator/metal nanocavity can act effectively as a single ENZ layer, where the cavity resonance corresponds to ENZ eigenmodes that can be excited by resonant tunneling[53]. Beyond a single layer, multiple layers of ENZ thin films offer a new degree of freedom to modulate optical coupling, while no work to our knowledge has investigated the interplay between near-field and far-field interactions[54–58].

Here we demonstrate an analogy of resonant quantum tunneling in optical systems based on ENZ multilayers, which enable long-range optical interactions beyond evanescent near-field coupling. Subwavelength enhanced electromagnetic fields and nearly-zero phase shift within the ENZ layers facilitate anti-correlated field intensity oscillations as a function of interlayer separation up to hundreds of microns, even with materials loss included, orders of magnitude longer than weak thin-film interference in dielectrics. To probe the separation-dependent optical responses, we deposited ITO/silica/ITO multilayers with a graded silica thickness using angled magnetron sputtering. Measured resonance intensity at ENZ wavelengths showed dependence on the dielectric spacer thickness. Furthermore, oscillatory SHG intensities were observed in ITO multilayers with an increased silica layer spacing. Optical excitation in resonance with the outer or inner ITO layer induces anti-correlated SHG intensities. Such a hybrid system that sustains both near-field enhancement and long-range radiative coupling offers intriguing prospects for large-scale integrated photonics and scalable quantum photonics.

## Results

Figure 1a depicts a summary plot to compare the near-field enhancement and optical interactions in different photonic systems. Plasmonic systems based on noble metals can support intense optical fields, and evanescent wave coupling is available at submicron scales. Compared to low-index dielectrics that exhibit weak thin-film interference, ENZ thin films manifest orders of magnitude stronger and localized electric fields confined by the ENZ/dielectric interfaces[49], which contributes to a significantly longer coupling distance. In the meantime, the relatively uniform phase distribution within ENZ thin films facilitates long-range radiative coupling between different layers. Figure 1b shows an analogy between quantum tunneling in electronics and long-range coupling in photonic systems based on ENZ materials. Similar to resonant tunneling through quantum double barriers[59], which is governed by constructive interference sensitive to relative phases of the wave function (see supplementary note), the confined evanescent waves within ENZ layers can couple with each other over long distances as tuned by the interlayer thickness.

We used magnetron sputtering to deposit ITO thin films with a controlled layer thickness and electron doping level. The fitted optical permittivity of ITO samples by ellipsometry followed a characteristic dispersion behavior of a Drude model that describes materials systems containing free electrons ($\varepsilon(\omega) = 1 - \frac{\omega_p^2}{\omega^2 + i\gamma\omega}$, where $\omega_p$ is plasma frequency, and $\gamma$ is damping coefficient) (Fig. 1c). At the plasma frequency ($\omega_p = \left(\frac{n_e e^2}{\varepsilon_0 m_e}\right)^{1/2}$, where $m_e$ is the effective mass and $n_e$ is the free electron density), ENZ condition was reached at $\lambda_{ENZ} = 1.9\ \mu m$ with a nearly-zero real part of permittivity. We further treated ITO thin films with post-growth rapid thermal annealing, which increases carrier concentration $n_e$ and hence blue shifts the ENZ wavelength. In the measured transmission spectrum under TM polarization, a resonance dip

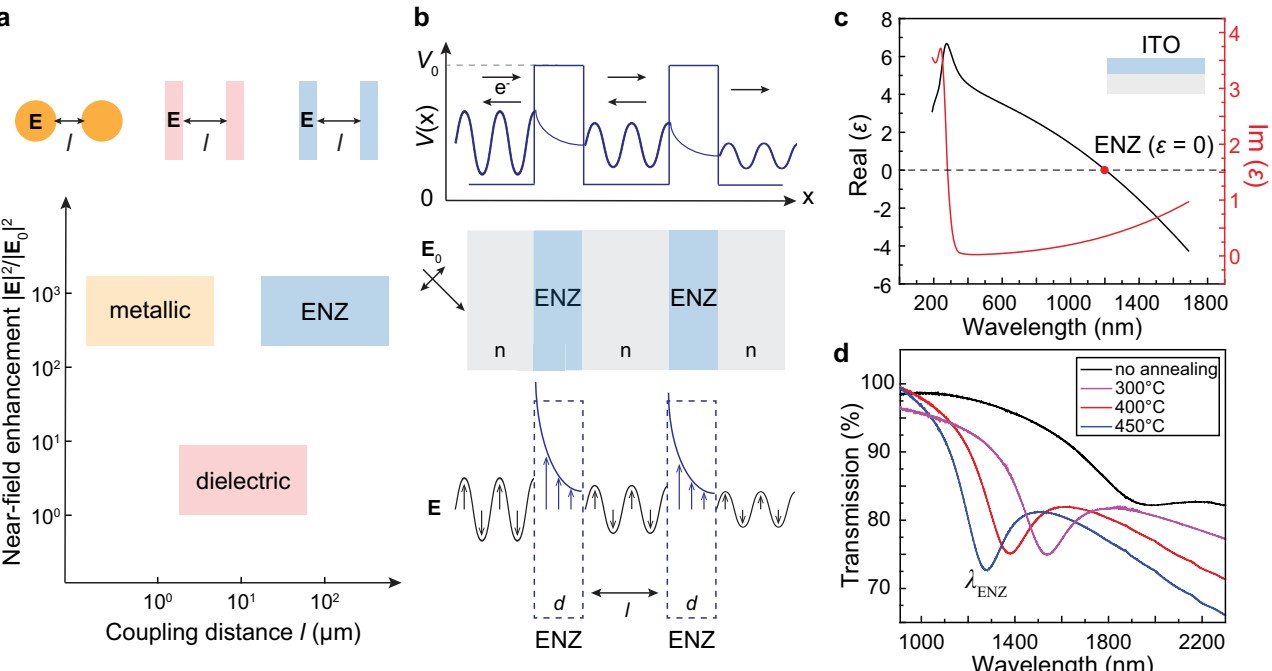

**Fig. 1 | Scheme of the long-range optical coupling in ENZ materials and the analogy to resonant quantum tunneling. a** Scheme for comparing optical coupling among the ENZ, metallic, and dielectric systems. **b** Scheme of resonant tunneling through quantum double barriers ($E < V_0$), and the analogy to an optical system composed of two ENZ thin films separated by a dielectric spacer. **c** Near-zero optical permittivity was observed close to the plasma frequency ($\lambda_{ENZ} = 1.2\ \mu m$) of an ITO thin film from Sigma Aldrich. Measured imaginary part of permittivity is $\varepsilon_i = 0.35$. **d** Measured transmission spectra of an ITO thin film (thickness $d = 50\ nm$) fabricated by magnetic sputtering. The ENZ wavelength was indicated by the dip position in the transmission plots measured by Fourier-transform infrared spectroscopy, which is tunable from $\lambda_{ENZ} = 1.25\ \mu m$ to $\lambda_{ENZ} = 1.55\ \mu m$ by thermal annealing after magnetic sputtering. The input light was under TM polarization.

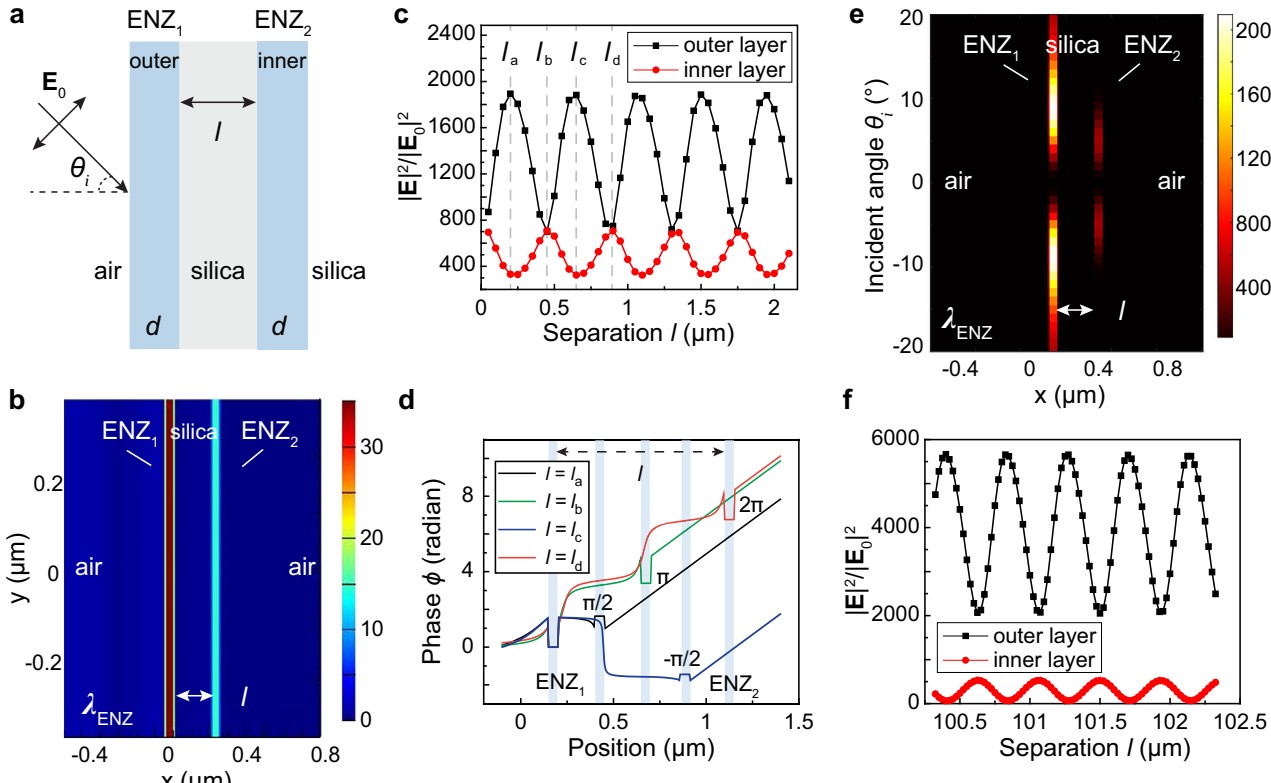

**Fig. 2 | Long-range optical coupling in ENZ multilayers and the anti-correlated optical near-field intensities. a** Schematic of two ENZ thin films separated by a silica spacer layer. **b** Near-field distribution $|\mathbf{E}|/|\mathbf{E}_0|$ of optical coupling between two ENZ layers with a separation $l = 0.2 \, \mu m$. **c** Oscillatory and anti-correlated optical field intensity in ENZ outer (left) and inner (right) layers on silica with an increased spacer separation. **d** Phase distribution plot for the ENZ/silica/ENZ multilayers at different interlayer separation $l$ in panel c. The shaded regions suggest location of the outer and inner ENZ layers. For separation distance $l = l_a$, $l_b$, $l_c$, and $l_d$, the phase difference between the inner and outer ENZ layer ($\Delta\phi> = \phi_{inner} - \phi_{outer}$) is $\pi/2$, $\pi$, $-\pi/2$, and $2\pi$, respectively. **e** Angle-dependent optical fields ($|\mathbf{E}|^2/|\mathbf{E}_0|^2$) of the two identical ENZ thin layers with a separation $l = 0.2 \, \mu m$ under TM polarization. **f** Anti-correlated optical near fields between ENZ double layers with a spacing up to hundreds of microns. The film thickness is $d = 50$ nm with $\varepsilon_i \cong 0$, pump wavelength $\lambda = 1.3 \, \mu m$ and incident angle $\theta_i = 7°$.

was observed at $\lambda_{ENZ}$, which can be tuned between 1.25 and 1.55 μm by varying the operation temperature over thermal annealing (Fig. 1d).

For a single-layer ENZ thin film at oblique incidence, the continuity of the electric displacement field **D** at the ENZ/dielectric interface ($\nabla \cdot \mathbf{D} = \rho_{e0}$) induces strongly enhanced optical fields inside the ENZ layer with characteristics of evanescent waves under TM polarization (Supplementary Movies 1-2). Based on finite-difference time-domain (FDTD) methods, an intense, sharp resonance at $\lambda_{ENZ}$ appeared in the transmission spectrum (Supplementary Fig. S1a, b). The optical extinction at ENZ wavelength originates from both light reflection and absorption, the latter indicating trapped light within the ENZ thin film as manifested by the near-field enhancement, which is attributed to Berreman[52] resonance (Supplementary Fig. S1c, d). The resonance intensity at the ENZ wavelength, as quantified by the depth of the transmission dip, decreased with increased materials loss (Supplementary Fig. S1e-f). In contrast, no resonance was available under transverse electric (TE) polarization or at normal incidence. Strong optical fields are confined at subwavelengths within the ENZ thin films (thickness $d = 50–200$ nm), and a higher $|\mathbf{E}|^2$ intensity was available for thinner ENZ films and those with better refractive index contrast at the materials interface (Supplementary Fig. 2).

In numerical modeling, long-distance optical coupling exists between two ENZ layers separated by a silica spacer (Fig. 2a). Despite the dielectric spacer, optical near fields are enhanced within both ENZ layers (Fig. 2b, Supplementary Movies 3-4). Under TM polarization, the electric fields within ENZ thin layers show intensity oscillations as a function of interlayer separation $l$, and $|\mathbf{E}|^2$ intensity is anti-correlated between the outer and inner ENZ layers (Fig. 2c). Such behavior can be

attributed to the varying optical path with increased separation $l$, which results in an optical phase shift that introduces either constructive or destructive interference. The phase plot in Fig. 2d suggests that ENZ thin films maintain nearly zero phase shift within a single layer, and the correlated phase difference between two films is determined by the separation distance.

As analogous to resonant tunneling through double barriers (see supplementary note), the inner ENZ layer's electric field intensity reaches a local maximum when constructive interference occurs between two layers with a phase difference of $m\pi$, where $m$ is an integer (Supplementary Fig. 3). Consistently, the oscillation period is sensitive to the refractive index of the spacer layer. Under TE polarization, weak optical near fields in ENZ multilayers showed no intensity correlation between each other (Supplementary Fig. 4). Different from the long-range coupling in ENZ layers, evanescent wave coupling at submicron scales exists in double metal thin films that sustain propagating surface plasmon polaritons (Supplementary Fig. 5). Similarly, we observed uncorrelated near fields between two clapping, low-index dielectric thin films, where the optical fields are orders of magnitude weaker and support only thin-film interference (Supplementary Fig. 6).

Optical field distributions suggest that the most intense optical fields within an ENZ thin film occur at a certain oblique incident angle (Fig. 2e), which originates from the angle-dependent field intensity at the materials interface and the angle-sensitive penetration depth of evanescent waves[49]. The anti-correlated field intensity oscillations are sustained for ENZ double layers with a wide separation up to hundreds of microns (Fig. 2f, Supplementary Figs. 7-8). Such an interaction distance is two orders of magnitude longer compared to the short

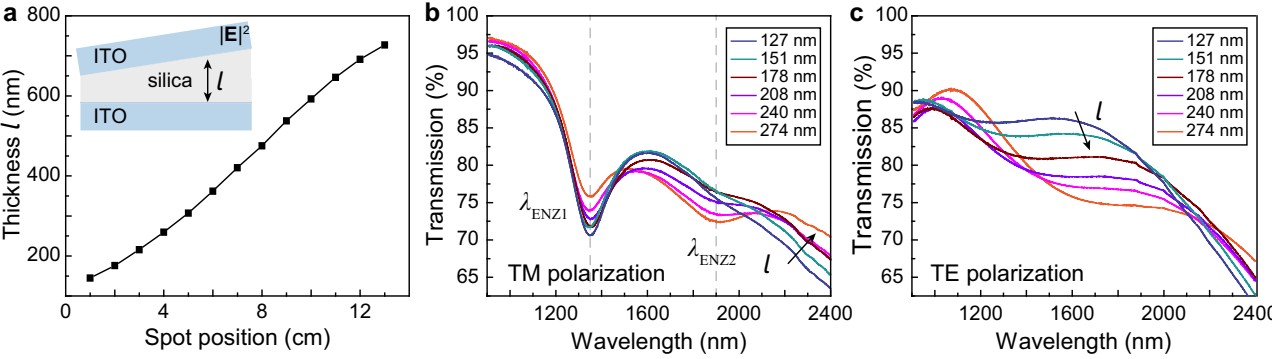

**Fig. 3 | ITO multilayers with graded spacer layer thickness and polarization-dependent linear optical properties. a** Thickness variation of the graded silica thin film across a 6 inch silica wafer as characterized by Fourier-transform infrared spectroscopy. Inset shows a scheme of ITO multilayers with a wedged silica spacer layer prepared by angled magnetron sputtering deposition. **b** Under TM polarization and at an incident angle of $\theta_i = 40°$, the measured resonance intensities at $\lambda_{ENZ1}$ and $\lambda_{ENZ2}$ show anti-correlated dependence on the increased silica spacer thickness $l$. **c** No transmission dip at ENZ wavelengths was observed under TE polarization.

penetration depth of evanescent waves within an ENZ material, suggesting long-range radiative optical coupling not accessible in conventional photonic systems. Long-range coupling with anti-correlated near fields is preserved for ENZ double layers with materials loss included (Supplementary Fig. 9). Furthermore, in multilayer ENZ thin films containing three distinct ENZ wavelengths (1.2, 1.1, and 1 μm, respectively), varying the pump wavelength can selectively address the submicron region of ENZ thin films to couple with each other (Supplementary Fig. 10). Hence, compared to electronic systems, optical wavelength can be a new degree of freedom in the spectral domain to determine the spatially resolved, subwavelength regions for interlayer coupling.

We fabricated ITO double-layer thin films with a wedged silica spacer to investigate the separation-dependent optical coupling. Based on a multi-step angled sputtering deposition process (Methods, Supplementary Fig. 11), where the sample holder is tilted to produce a varied distance between different sites of the substrate and the silicon sputtering target, we achieved a graded silica film with thickness varying from $d = 145$ nm to 727 nm across a 6 inch silica wafer (Fig. 3a). Under TM polarization, the resonance intensity indicated by the depth of a transmission dip at $\lambda_{ENZ1}$ and $\lambda_{ENZ2}$ showed anti-correlated dependence on the dielectric spacer thickness (Fig. 3b). As the spacer thickness increased from $l = 127$ to 274 nm, the resonance intensity at $\lambda_{ENZ1}$ continuously decreased, while the resonance intensity at $\lambda_{ENZ2}$ increased. In contrast, we observed no transmission dips for ITO multilayers under TE polarization, and the background transmission intensity decreased at longer wavelengths because of the increased materials loss (Fig. 3c). FDTD modeling on ENZ multilayers showed oscillatory behavior of resonance intensity as a function of dielectric spacer thickness, where the oscillation period is defined by the pump wavelength, incident angle, and refractive index of the surrounding layers (Supplementary Figs. 12a–c). The modeled resonance intensity, as quantified by the depth of the transmission dip, continuously decreased with an increased spacer separation at $\lambda_{ENZ1}$ and increased instead at $\lambda_{ENZ2}$ (Supplementary Figs. 12d–h), which agrees well with the measured transmission spectra evolution in Fig. 3b. Such anti-correlated resonance intensity change is consistent with the electric field oscillations in Fig. 2c, where the first half of the oscillation period was captured.

Besides linear optical properties, long-range coupling between ENZ layers can induce separation-dependent nonlinear optical behaviors. Enabled by the enhanced electromagnetic fields confined within ENZ thin films, SHG signals whose intensities scale with $|E|^4$ can be generated and were used to probe the change of near-field intensities in ITO single and multilayers. A short-pulsed femtosecond laser excited the ITO thin films at oblique incidence (Supplementary Fig. 13). For a single-layer ITO ($\lambda_{ENZ} = 1.2$ μm) under TM polarization, intense optical near fields at ENZ wavelengths contribute to strong SHG signals. We observed an SHG emission peak at 619 nm with a pump wavelength at $\lambda = 1.25$ μm (Supplementary Fig. 14a). The power-dependent output curves showed that the SHG peaks have a power-law rising slope of 1.7, indicating a two-photon nonlinear process. Note that a slight deviation from a rising slope of 2 can be attributed to additional optical scattering signals collected by the CCD camera. In contrast, only one sideband at 650 nm was observed under TE polarization, and its linear rise with increased pump power indicates the origin from optical scattering in the light path (Supplementary Fig. 14b). We observed almost no SHG signals from a glass substrate, which suggests that the detected SHG signals come from ITO thin films (Supplementary Fig. 14c). Wavelength-dependent nonlinear measurements showed a maximum SHG intensity when the pump wavelength was close to $\lambda_{ENZ}$ (Supplementary Fig. 15).

To probe the optical near fields within ITO, we first investigated a single-layer ITO film with graded thickness variation across the sample based on angled sputtering deposition (Methods, Fig. 4a). FDTD methods were used in the SHG modeling, where a nonlinear material was integrated into a base ENZ matrix defined by the Drude model to simulate the nonlinear optical responses. For a single ENZ layer with a thickness $d = 50$ nm, SHG signals appeared at 600 nm with a pump wavelength $\lambda = 1.2$ μm at TM polarization under oblique incidence (Supplementary Figs. 16a–c). Increased film thickness led to weaker optical fields within the ENZ thin film and hence continuously decreased SHG intensities (Fig. 4b). Varying pump wavelengths induced different SHG wavelengths, and no nonlinear responses exist for a pump at normal incidence (Supplementary Figs. 16d, e).

In the graded single ITO film, a post-annealing treatment at 450 °C introduced a redshift of $\lambda_{ENZ}$ from 1.2 to 1.45 μm characterized by ellipsometry as the film thickness increased from $d = 72$ nm to $d = 163$ nm across the sample (Supplementary Fig. 17), which could come from grain size difference for varying film thickness over thermal annealing[60]. With a pump wavelength at $\lambda_{pump} = 1.2$ μm that equals $\lambda_{ENZ}$ at $d = 72$ nm, we scanned over the graded ITO film. Measured SHG signals at 600 nm showed a maximum intensity for film thickness $d = 72$ nm, where a thinner ITO film produced a stronger optical near field as the pump wavelength is in better resonance with $\lambda_{ENZ}$ (Fig. 4c, black curve). In contrast, at a pump wavelength $\lambda_{pump} = 1.5$ μm (close to $\lambda_{ENZ}$ of $d = 163$ nm), a maximum intensity for SHG signals at 750 nm was observed for film thickness $d = 116$ nm instead (red curve), which is a tradeoff between a thinner ITO film for stronger optical fields and a better spectral match between $\lambda_{ENZ}$ and the pump wavelength $\lambda_{pump}$.

We measured SHG responses from wedged ITO multilayers to probe the separation-dependent optical near-field intensities. Similar

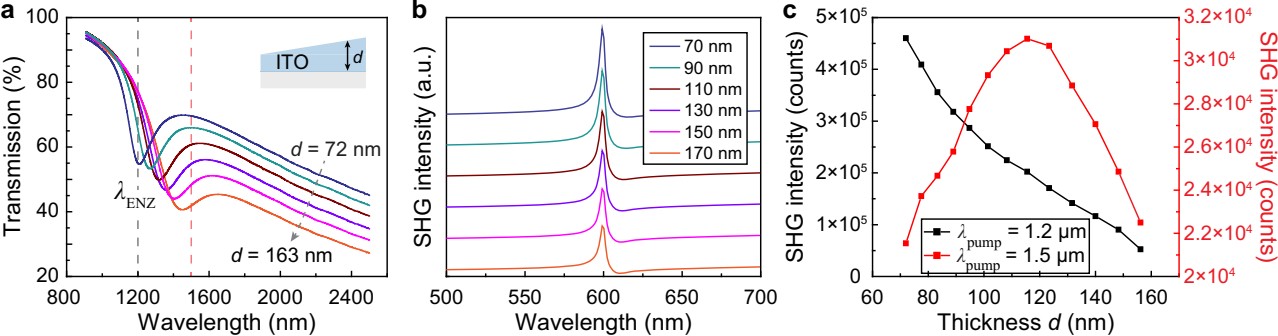

**Fig. 4 | Second harmonic generation from a graded single ITO thin film.**
**a** Measured shifted ENZ wavelength and intensity for a single-layer ITO thin film
with thickness varied from $d = 72$ to 163 nm. **b** Modeled SHG emission from a single
ITO layer with a thickness ranging from $d = 70$ to 170 nm. A uniform ENZ wave-
length is set at $\lambda_{ENZ} = 1.2\ \mu m$ in the modeling with incident angle $\theta_i = 5°$. SHG
curves were offset from top to bottom. **c** Measured SHG intensities at the doubled fre-
quencies. The pump wavelength is $\lambda = 1.2\ \mu m$ (close to $\lambda_{ENZ}$ at $d = 72$ nm, black
curve) with a power density of 636 W/cm², and $\lambda = 1.5\ \mu m$ (close to $\lambda_{ENZ}$ at
$d = 163$ nm, red curve) with a power density of 95 W/cm², respectively, corre-
sponding to the dashed lines in panel a.

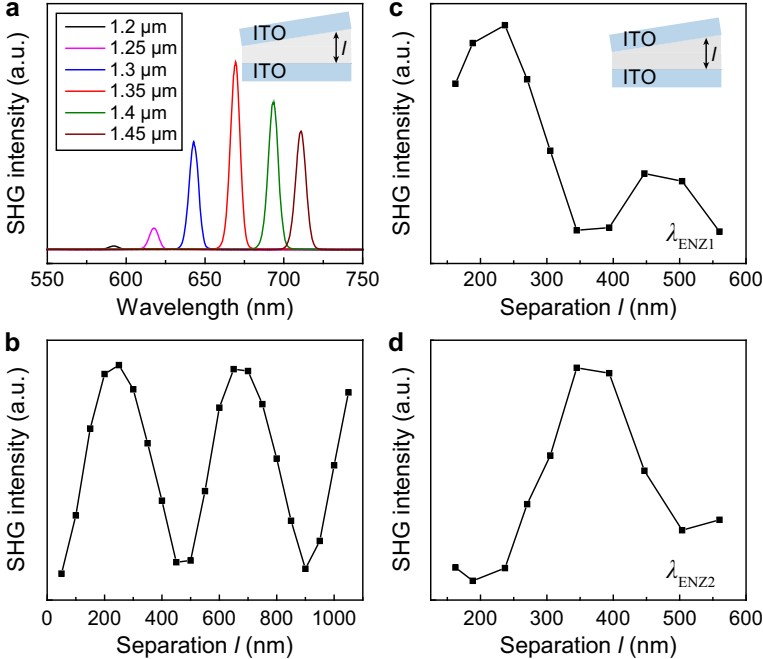

**Fig. 5 | Second harmonic generation to probe the oscillatory and anti-
correlated optical near-field intensities in ITO multilayers.** **a** Measured SHG
signals with a scanned incident wavelength $\lambda_{pump}$ from 1.2 to 1.44 $\mu m$. SHG was
measured from wedged ITO multilayers (film thickness $d = 50$ nm) at a separation
of $l = 178$ nm. **b** Modeled SHG intensities from the outer ENZ layer of a wedged ENZ/
SiO₂/ENZ multilayer at a pump wavelength $\lambda = 1.3\ \mu m$. Incident angle was at $\theta_i = 7°$.
**c** Measured oscillatory, anti-correlated second-harmonic responses with the pump
wavelength at 1.4 $\mu m$ with a power density of 541 W/cm² and, **d** 1.7 $\mu m$ with a power
density of 73 W/cm², respectively.

to a single-layer ITO film, by varying the pump wavelength, we found
that the strongest SHG appeared in ITO multilayers when $\lambda_{pump}$ was
tuned close to $\lambda_{ENZ} = 1.35\ \mu m$ (Fig. 5a). A point monitor positioned at
the center of the outer ENZ layer in FDTD modeling showed an
oscillatory change in the SHG intensity as a function of interlayer
separation $l$ (Fig. 5b). By pumping at $\lambda = 1.4\ \mu m$, which was close to
$\lambda_{ENZ}$ of the bottom ITO layer, we scanned over the wedged ITO
multilayers in experiment and observed oscillatory SHG intensities as
$l$ increased from 162 to 560 nm (Fig. 5c). The pump wavelength was
also tuned to $\lambda = 1.7\ \mu m$ to probe the field change of the top ITO layer
($\lambda_{ENZ} = 1.9\ \mu m$, Fig. 5d). Interestingly, SHG intensity showed an anti-
correlated oscillation behavior with that at $\lambda = 1.4\ \mu m$, which is con-
sistent with the anti-correlated electric fields between the outer and
inner ENZ layers (Fig. 2c). Note that compared to the modeled SHG
intensity, the difference in measured oscillation period can be
attributed to additional interference at the ITO/silica/ITO interfaces
(Supplementary Fig. 18).

In summary, we achieved long-range optical interactions in ENZ
multilayers up to hundreds of microns beyond evanescent wave cou-
pling in the near field. This system shows an analogy to resonant
tunneling through quantum double barriers in terms of the oscillatory
change in electromagnetic field intensity enabled by constructive
interference. Both transmission and SHG measurements revealed the
anti-correlated, intense optical near fields within two ENZ layers.
Inspired by these findings, we anticipate that ENZ multilayers may
serve as an optical ruler with a deep subwavelength spatial resolution
to sense optical and biomedical environments. Replacing ITO materi-
als with rationally designed photonic crystals[36] allows for achieving
effectively zero permittivity with lower material loss, which opens new
prospects in nonlinear optics. Such a system offers a new strategy for

modulating long-range optical interactions while maintaining a mode footprint at submicron scales, which can benefit scalable solid-state photonics, enhanced nonlinear optics, remote sensing, and quantum photonics.

## Methods

### Fabrication of wedged ITO multilayers

We fabricated ITO double-layer thin films with a wedged silica spacer based on a multi-step sputtering process. A thin-layer ITO film was first deposited on a fused silica substrate based on magnetron sputtering (ast-sputter). Rapid thermal annealing (350 °C, 1 min) was used to blue shift the ENZ wavelength to around $\lambda_{ENZ} = 1.3\,\mu m$. For reactive sputtering with a silicon target, the sample holder is angled during the deposition, which produces a silica thin film with a graded thickness variation. A second layer of ITO was then sputtered on top to form the ITO/silica/ITO multilayers. With a sputtering power of 100 W, Ar gas flow at 60 sccm, and deposition time of up to 3 h, the deposited $SiO_2$ layer showed a thickness variation from 100 nm up to 800 nm from one edge to another.

### Fourier-transform infrared spectroscopy measurements

Fourier-transform Infrared Spectroscopy (Nocolet Continuum iS50 FT-IR microscope, Thermo Scientific) was used to quantify the separation-dependent transmission spectra in ITO double layers at room temperature. The detection range was between 4000 and 11000 cm$^{-1}$ and the pump spot was at 1 cm in diameter. A Glan-Thompson crystal polarizer was placed in front with an operation wavelength between 400–2300 nm (Thorlabs). The sample was mounted on a rotational stage (Thorlabs) with a one-axis translation mount (Thorlabs, XF100). Measurements were conducted at room temperature based on a DTGS KBr detector. An incident angle between 0°–50° was realized by manually rotating the sample stage.

### SHG measurements

A Yb-based femtosecond laser (PHAROS, Light Conversion) and an optical parametric amplifier (ORPHEUS, Light Conversion) were used for the optical pump and characterization at room temperature. The pulse duration was 200 fs, and the repetition rate was set to 150 kHz. The ultrafast pulsed laser supports a tunable output wavelength between 0.8–2 μm. The laser beam was focused by an optical lens (focal distance $f = 3$ cm) on the ITO samples with a 20-μm diameter circular spot at an incident angle of 45°. The average pump power density for SHG measurements varied from 64 to 636 W/cm². The emission signals at 45° were recollected by a focal lens ($f = 5$ cm) before entering a Si-based CCD camera (Princeton Instruments, Acton Series SP-2300i). The condition of oblique incidence, TM polarization, and a short-pulsed -fs laser is critical for exciting SHG signals in ITO thin films.

### FDTD modeling

FDTD calculations with commercial software (FDTD solution, Lumerical Inc., Vancouver, Canada) were used to model the linear optical properties and nonlinear responses of ENZ multilayer thin films. We used a uniform mesh size of <5 nm ($x$, $y$, and $z$) for the accuracy of electromagnetic field calculations within the ENZ regions. A nonlinear material model (chi2 material) with a second-order nonlinear coefficient of $\chi = 3 \times 10^{-10}$ m/V was used to model the nonlinear responses, where a Drude-based material serves as the host material matrix.

## Data availability

The data supporting the findings of this study have been included in the main text and Supplementary Information. All other relevant data supporting the findings of this study are available from the corresponding authors upon request.

## Code availability

The codes used for this study are available from the corresponding authors upon request.

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

## Acknowledgements

D.W. was supported by Shanghai Category IV Peak Discipline Funding, Fudan University Start-up Funding, and the Miller research fellowship at Miller Institute, University of California, Berkeley. The thin film deposition and optical measurement part of this work was funded by the U.S. Department of Energy, Office of Science, Office of Basic Energy Sciences, Materials Sciences and Engineering Division under contract DE-AC02-05-CH11231 (EMAT program KC1201). The second harmonic generation measurement part was supported by U.S. Department of Energy, Office of Science, Basic Energy Sciences, Materials Sciences and Engineering Division under Contract No. DE-AC02-05-CH11231 within the van der Waals heterostructure program KCFW16. P.C. was supported by the National Natural Science Foundation of China (No. 12304540).

## Author contributions

D.W. and J.W. conceived the idea of optical interactions based on ENZ materials. D.W. fabricated the ITO single and multilayer films with magnetron sputtering (R.G.) and carried out measurements on linear optical properties with Fourier-transform infrared spectroscopy (N.G., J.L., Q.W., and A.J.) and ellipsometry (S.W. and J.Y.), as well as thin-film thickness characterization with atomic force microscopy (P.C.). D.W. and Z.L. performed the nonlinear optics measurements (F.W.). D.W. performed numerical simulations of the ENZ multilayer long-range coupling, near-field distributions, and SHG enhancement. D.W. and J.W. analyzed the data and wrote the manuscript. All authors commented on and revised the manuscript.

## Competing interests

The authors declare no competing interests.
