## [Transparent Peer Review file · Nature Communications]

Long-range Optical Coupling with Epsilon-near-zero Materials

Corresponding Author: Dr Danqing Wang

Version 0:

Reviewer comments:

Reviewer #1

(Remarks to the Author)

The authors demonstrate long-range optical coupling in optical systems based on ENZ materials, analogous to resonant quantum tunneling in electronic systems. This long-range optical interaction transcends conventional evanescent field coupling, offering new possibilities for applications in optical communication, large-scale photonic circuits, and hybrid quantum photonic systems. The idea is interesting and it implies a potential breakthrough in both zero-index photonics and quantum physics. I have the following questions/comments:

1. As for the analogy of ENZ tunneling and quantum tunneling, the quantum barrier is usually regarded as negative-epsilon material because they support evanescent waves. Here ENZ material is used instead. But ENZ materials have propagating waves inside, although they have huge impedance mismatch. So, is there any fundamental difference between the two schemes?
2. What is the influence of the imaginary part of ITO? I mean, how does the performance change with the imaginary part?
3. ENZ materials provide better long-range optical coupling. But how is the transmission related to coupling efficiency? Why does the transmission dip emerge at ENZ wavelength?
4. The manuscript mentions the differences between ENZ materials and other materials (such as metals) in terms of optical coupling, but it seems a more detailed comparative analysis is lacking. It is thus recommended to add some results about the case of traditional evanescent near-field coupling to demonstrate the advantages of ENZ materials in coupling distance, field enhancement.
5. Metamaterials or photonic crystals can realize effective permittivity with very low imaginary part. So, is it possible to use metamaterials or photonic crystals to replace the ITO material used here? Pls. clarify.
6. There are some typos, e.g. for the equation of the Drude model $\epsilon(\omega)=\dots$, the angular frequency should be written as " ω " instead of " w ".
7. I recommend the authors to include some literature related to ENZ materials or zero-index materials such that the readers have a more comprehensive picture of the ways to realize tunneling with zero-refractive-index, e.g. *Advanced Optical Materials*, 12, 2401569 (2024), *Nanophotonics* 12, 2007 (2023), *Physical Review Letters* 127, 123902 (2021), *Physical Review Letters* 124, 074501 (2020), *Light: Science and Applications* 7, 50 (2018), *Laser & Photonics Reviews* 12, 1800001 (2018), *Physical Review X* 8, 031035 (2018), etc.

Reviewer #2

(Remarks to the Author)

[Editorial Note: Reviewer #2's attachment has been appended to the end of this file]

Reviewer #3

(Remarks to the Author)

The manuscript entitled "Long-range Optical Coupling with Epsilon-near-zero Materials" by Wang et al. presents a compelling investigation into a multilayered structure (ENZ/Spacer/ENZ), where anti-correlated electromagnetic field intensities are explored through second harmonic generation (SHG) from wedged indium tin oxide (ITO) multilayers. While the study addresses an interesting and timely topic, I believe significant revisions are necessary before the work can be

considered for publication.

Below, I outline my detailed comments and suggestions that the authors should address in a revised version of the manuscript:

- **Quantum Mechanical Analogy:** The authors are encouraged to consider the following reference in their discussion of the quantum mechanical analogy, particularly when addressing resonant tunneling effects in ENZ structures:
Caligiuri, Vincenzo, et al. "A semi-classical view on epsilon-near-zero resonant tunneling modes in metal/insulator/metal nanocavities." *Nano Letters* 19.5 (2019): 3151–3160.
The novelty of the proposed approach should be clarified in light of the cited work.
- **Comparison with Recent Studies:** The manuscript reports numerical and experimental studies on ENZ/spacer/ENZ configurations with varied spacer thicknesses. The authors should also discuss the following recent contribution and clarify how their work differs or advances the field:
Lio, Giuseppe Emanuele, et al. "Unlocking Optical Coupling Tunability in Epsilon-Near-Zero Metamaterials Through Liquid Crystal Nanocavities." *Advanced Optical Materials* 12.13 (2024): 2302483.
- **Material Choice Justification:** The rationale behind using an ITO/SiO₂/ITO structure instead of other alternatives, such as Ag/ITO/Ag, should be discussed. The latter may offer advantages in tuning the resonance without requiring annealing processes.
- **Field Confinement at Resonance:** On page 4, the authors are requested to provide numerical simulations showing the electric field confinement within the SiO₂ layer at resonance and off-resonance conditions to support their interpretation.
- **Clarification of Terminology:** On page 5, lines 98–101, the expression "electric field displacement" is vague and should be clarified to avoid ambiguity.
- **Reference to Supplementary Material:** On page 5, line 101, the transmittance spectrum mentioned—if referring to that shown in Supplementary Figure S1—should include a direct reference (e.g., "see Supplementary Fig. 1").
- **Resonance at Normal Incidence:** The manuscript should explicitly explain why the proposed system does not exhibit any resonance at normal incidence. This is a fundamental aspect that warrants discussion in the main text.
- **Clarity of Figures:** Figures 2b and 2e lack clarity in depicting the structure. The boundaries of the ENZ layers, the spacer, and the surrounding medium (e.g., air) are not clearly delineated. A clearer schematic is recommended.
- **Electric Field Maps:** To enhance understanding of light confinement, the authors are encouraged to include electric field distribution maps corresponding to the configuration in Figure 3, particularly for selected incident angles at both λ_{ENZ1} and λ_{ENZ2} .
- **Stylistic Suggestion:** The sentence spanning lines 172–174 on page 9 appears redundant and may be merged with the preceding statement for improved readability and conciseness.
- **SHG Response Interpretation:** In Figure 4b, the SHG peaks appear unshifted even when the cavity thickness is altered, despite pumping at λ_{ENZ1} . This seems inconsistent with the observations in Figure 5a, where SHG peaks shift. While a reduction in intensity is acknowledged, a wavelength shift might also be expected. This discrepancy should be discussed in greater detail.
- **Conclusion Revision:** The conclusions section presently overstates the applicability of the proposed structure to quantum applications. A more balanced summary focusing on the actual findings would improve the overall credibility and focus of the manuscript.
- **In Methods section,** particularly concerning SHG, the authors should specify the pulsed laser power used and the corresponding beam size, to ensure reproducibility and proper comparison with other studies.

Reviewer #4

(Remarks to the Author)

This paper investigates long-range optical interactions in epsilon-near-zero (ENZ) materials, drawing a compelling analogy to quantum tunneling in electronic systems. The most significant findings can be summarized as follows:

- (a) The electric field intensities within ENZ layers exhibit anti-correlated oscillations as the interlayer separation varies;
- (b) This behavior is probed through second harmonic generation (SHG), a nonlinear optical process in which two photons combine to emit a single photon at twice the original frequency.

These results are both relevant and well-supported by robust experimental data and numerical simulations. In my opinion, the manuscript is highly appropriate for the audience of *Nature Communications*, and I recommend acceptance after addressing the following minor point.

To interpret the observed optical behavior, you propose an analogy with resonant quantum tunneling through a double-barrier system. While this comparison is insightful, I believe it is not entirely novel, as a similar perspective was already presented in the following reference:

Caligiuri, V., Palei, M., Biffi, G., Artyukhin, S., & Krahne, R. (2019). A semi-classical view on epsilon-near-zero resonant tunneling modes in metal/insulator/metal nanocavities. *Nano Letters*, 19(5), 3151–3160.

It would strengthen the manuscript to more clearly highlight both the similarities and the novel aspects of your approach in relation to this previous work. If relevant, clarifying how your results expand on or differ from the viewpoint proposed by Caligiuri et al. could further enhance the contribution of your paper.

Version 1:

Reviewer comments:

Reviewer #1

(Remarks to the Author)

The authors have carefully revised their manuscript according to the reviewers' comments. My questions have been well answered in their reply. I thus recommend publication of this work on Nature Communications.

Reviewer #2

(Remarks to the Author)

In this manuscript, Wang et al., show the analogue of long-range resonant electron quantum tunnelling in optical systems composed of epsilon-near-zero (ENZ) thin films. The authors have given a point-by-point response to my previous review comments.

While I am satisfied with the response provided by the authors to most of my previous queries, there is a concern regarding one of the aspects. In response to point#3 of comment#8, the authors had written that the transmission dips at ENZ wavelengths in experimental data is smaller than in the simulation due to the material loss of the ITO thin film. While I agree with this, can the authors not perform the simulation by taking into account the material loss? They do have the experimentally measured the optical constants of ITO film (Figure S17). Since the modulation of transmission dips is a crucial piece of data that supports the assertion of this work, I feel the authors should redo the simulation with the measured optical constants of ITO which I would expect should give a smaller depth of modulation consistent with the experimental data.

Reviewer #3

(Remarks to the Author)

The Authors addressed all my comments and doubts. I appreciate their effort and the presented manuscript. I support its publication in Nature Communications.

Reviewer #4

(Remarks to the Author)

Dear Authors,

I notice that my concern has been effectively and convincingly addressed. I have no more issues with the manuscript and it is worth publication in Nature Communications.

Kind regards

REVIEWER # 1:

Comment 1: “The authors demonstrate long-range optical coupling in optical systems based on ENZ materials, analogous to resonant quantum tunneling in electronic systems. This long-range optical interaction transcends conventional evanescent field coupling, offering new possibilities for applications in optical communication, large-scale photonic circuits, and hybrid quantum photonic systems. **The idea is interesting and it implies a potential breakthrough in both zero-index photonics and quantum physics.**”

Our response: We thank the reviewer for taking the time to review our manuscript and for appreciating the importance of our findings.

Comment 2: “As for the analogy of ENZ tunneling and quantum tunneling, the quantum barrier is usually regarded as negative-epsilon material because they support evanescent waves. Here ENZ material is used instead. But ENZ materials have propagating waves inside, although they have huge impedance mismatch. So, is there any fundamental difference between the two schemes?”

Our response: We thank the reviewer for their question regarding the difference between metals and ENZ in the quantum tunneling scheme.

First, we note that in our system under oblique-angle excitation, where the total internal reflection condition is at the air/ENZ interface, evanescent waves instead of propagating waves exist within the ENZ materials. **New** light propagation videos were added as supplementary information for illustration.

The difference between ENZ materials and metals in the analogy to quantum tunneling includes:

- (1) Compared to metals, a unique and uniform phase distribution was maintained within ENZ;
- (2) Stronger near fields inside ENZ thin films compared to the metals.

These two factors both facilitate a long coupling distance between ENZ thin films up to hundreds of microns. In contrast, based on FDTD simulations, metal thin films showed a coupling distance only at the microscale, the same order as the wavelength of the input light (**Figure S5**). In addition, an ENZ layer can

New Figure S7. The electric field distribution in ENZ thin films and the silica spacer layer.

serve as a quantum barrier, while the metal/insulator/metal multilayers together serve as a single barrier in the quantum tunneling analogy [*Nano Letter* 19, 5, 3151–3160 (2019)].

Changes to manuscript: We added new modeling results and related discussions to the manuscript (**new** Figure S7). We added new simulation videos of light propagation to the supporting information (**new** Supplementary Movies 1-4).

Comment 3: “What is the influence of the imaginary part of ITO? I mean, how does the performance change with the imaginary part?”

Our response: We thank the reviewer for this comment.

For a single ITO layer, the resonance intensity and the optical field intensity confined within the ITO layer decreased with increased material loss. We added the modeling results in the **new** Figure S1.

New Figure S1. Influence of materials loss to the ENZ resonance lineshape.

In addition, we studied the influence of different imaginary values of epsilon on the interlayer coupling distance. Results in Figure S9 suggest that long-range coupling with anti-correlated near fields is preserved for ENZ double layers up to 300 μm, even with material loss $\epsilon_i = 0.1$ included. Note that such

Figure S9. Influence of different imaginary part of epsilon to the long-distance mode coupling.

a loss value ($\varepsilon_i = 0.1$) is comparable to state-of-the-art transparent semiconductor oxides fabricated by epitaxial growth [*Nature Photonics* 11, 149–158 (2017)]. Therefore, although the near-field intensity decreased with increased materials loss, the long-distance coupling is robust with a coupling distance of hundreds of microns.

Changes to manuscript: We added additional modeling results and discussions to the new Figure S1.

“The resonance intensity at the ENZ wavelength, as quantified by the depth of the transmission dip, decreased with increased materials loss (Supplementary Fig. S1e-f).”

Comment 4: “ENZ materials provide better long-range optical coupling. But how is the transmission related to coupling efficiency? Why does the transmission dip emerge at ENZ wavelength?”

Our response: We thank the reviewer for this critical comment.

In Figure 1d, transmission dips indicate the percentage of light extinction, which partly comes from reflection (35%) and partly from absorption (65%). The absorbed light is trapped within ENZ thin films, and hence, optical near fields were strongly enhanced compared to the input light.

Therefore, as the optical permittivity approaches zero at the ENZ wavelength, the maximum near field enhancement happens, which corresponds to the same spectral position as the transmission dip. In addition, since the coupling strength scales with the optical field intensity, strongly enhanced near fields at the ENZ wavelength facilitate the long-distance coupling between ENZ layers.

Figure S1. Origin of the transmission dip at the ENZ wavelength.

Changes to manuscript: We added new discussions to the main text and supplementary information.

“The optical extinction at ENZ wavelength originates from both light reflection and absorption, the latter indicating trapped light within the ENZ thin film as manifested by the near-field enhancement, which is attributed to Berreman⁵² resonance (Supplementary Fig. S1c-d).”

Comment 5: “The manuscript mentions the differences between ENZ materials and other materials (such as metals) in terms of optical coupling, but it seems a more detailed comparative analysis is lacking. It is thus recommended to add some results about the case of traditional evanescent near-field coupling to demonstrate the advantages of ENZ materials in coupling distance, field enhancement.”

Our response: We thank the reviewer for this comment.

We have compared the long-range optical coupling in the ENZ system to the traditional evanescent near-field coupling in the following cases:

1. metals/air/metal multilayers

Different from the long-range coupling in ENZ layers, evanescent wave coupling exists in double metal thin films, which sustain propagating surface plasmon polaritons (Figure S5). Notably, the near-field enhancement is one order of magnitude weaker compared to the ENZ double layers. Evanescent near-field coupling is sustained at submicron scales for pumping at visible wavelengths.

Figure S5. Evanescent near-field coupling limited to submicron in double metal thin films.

2. air/silica/air multilayers

Another case for comparison is the air/silica/air multilayer system, where the near fields are enhanced within the low-index air thin films (Figure S6). Compared to ENZ double layers, uncorrelated near fields were observed between two low-index dielectric layers, where the optical fields are three orders of magnitude weaker. The weak thin-film interferences cannot sustain optical coupling with correlated near fields over distances, such as a separation $l = 10 \mu\text{m}$ and $l = 100 \mu\text{m}$ in the figure below. The summarized plot, including both near-field enhancement and coupling distance among different systems, is demonstrated in Figure 1a.

New Figure S6. Comparison of near field oscillations to the air/silica/air multilayer system.

Changes to manuscript: We added more discussions and a new figure to the manuscript (**new** figure caption of Figure S5, **new** Figure S6 and caption).

“Different from ENZ thin films, evanescent wave coupling exists in double metal thin films, which sustain propagating surface plasmon polaritons. No correlated optical intensity oscillation was observed in double Ag thin films as a function of interlayer separation l . The near field enhancement is one order of magnitude weaker compared to the ENZ double layers. The coupling distance at submicron scales is also much shorter than the ENZ thin films.”

“We compared the ENZ multilayer system to both metal and dielectric thin films that sustain leaky modes. Neither of these two cases showed correlated optical near fields, and both exhibited orders of magnitude shorter coupling distances compared to the ENZ multilayers.”

Comment 6: “Metamaterials or photonic crystals can realize effective permittivity with very low imaginary part. So, is it possible to use metamaterials or photonic crystals to replace the ITO material used here? Pls. clarify.”

Our response: We thank the reviewer for making this insightful point.

We have included the literature on exploiting metamaterials and photonic crystals for achieving ENZ materials. As noted by the reviewer, metamaterials or photonic crystals can realize effectively zero permittivity through engineering the mode degeneracy in the band structure [*Nature Materials* **10**, 582–586 (2011)]. The relatively low imaginary part in an all-dielectric system will be favorable for nonlinear optical responses. Such a system can, in principle, replace the ITO materials in the multilayers, and open future prospects for study.

Changes to manuscript: We added new references and related discussions to the conclusion paragraph (page 12).

“Replacing ITO materials with rationally designed photonic crystals [*Nature Materials* **10**, 582–586 (2011)] allows for achieving effectively zero permittivity with lower materials loss, which opens new prospects in nonlinear optics.”

Comment 7: “There are some typos, e.g. for the equation of the Drude model $\epsilon(\omega)=\dots$, the angular frequency should be written as “ ω ” instead of “ w ”.”

Our response: We thank the reviewer for this correction.

Change to manuscript: We corrected the typo in the main text (page 5) as follows:

“..... containing free electrons ($\epsilon(\omega) = 1 - \frac{\omega_p^2}{\omega^2 + i\gamma\omega}$, where ω_p is plasma frequency and γ is the damping coefficient). At the plasma frequency ($\omega_p = \left(\frac{n_e e^2}{\epsilon_0 m_e}\right)^{1/2}$ ”.

Comment 8: “I recommend the authors to include some literature related to ENZ materials or zero-index materials such that the readers have a more comprehensive picture of the ways to realize tunneling with zero-refractive-index, e.g. *Advanced Optical Materials*, **12**, 2401569 (2024), *Nanophotonics* **12**, 2007 (2023), *Physical Review Letters* **127**, 123902 (2021), *Physical Review Letters* **124**, 074501 (2020), *Light: Science and Applications* **7**, 50 (2018), *Laser & Photonics Reviews* **12**, 1800001 (2018), *Physical Review X* **8**, 031035 (2018), etc..”

Our response: We thank the reviewer for suggesting these new references on ENZ or zero-index materials as related to our manuscript. These papers show nice applications of ENZ materials in light tunneling, optical cloaking, perfect absorption, etc.

Change to manuscript: We have added the related new references and discussions to the introduction section in the main text (page 2, **new** references 29, 34, 35, 38).

REVIEWER # 2:

Comment 1: “In this manuscript, Wang et al., show the analogue of long-range resonant electron quantum tunnelling in optical systems composed of epsilon-near-zero (ENZ) thin films. While the proposition is interesting, in my view, the results do not fully attest to this and therefore, the manuscript is not yet at the level for publication in Nature Communications.”

Our response: We thank the reviewer for taking the time to review our manuscript. We provided point-to-point responses to the comments as below.

Comment 2: “There is a mismatch between data in Figs. 1c and 1d. In Fig. 1c, the ENZ point is at $\lambda = 1.2 \mu\text{m}$ whereas in Fig. 1d, the dip in transmission that corresponds to the ENZ point occurs at $\lambda = 1.9 \mu\text{m}$ (line 92 of manuscript text). This needs to be clarified.”

Our response: We thank the reviewer for this point.

Figure 1c corresponds to a typical permittivity plot of a typical, state-of-the-art ENZ thin film purchased from Sigma Aldrich (thickness $\sim 50 \text{ nm}$, surface resistivity $30\text{-}60 \Omega/\text{sq}$). In contrast, Figure 1d shows the transmission spectra of our ITO thin film samples in the lab as fabricated by magnetic sputtering with an ITO target.

Change to manuscript: We added more clarifications to the manuscript (**new** Figure 1 caption).

“Near-zero optical permittivity was observed close to the plasma frequency ($\lambda_{\text{ENZ}} = 1.2 \mu\text{m}$) of an ITO thin film from Sigma Aldrich.”

“Measured transmission spectra of an ITO thin film (thickness $d = 50 \text{ nm}$) fabricated by magnetic sputtering. The ENZ wavelength was indicated by the dip position in transmission plots with Fourier-transform infrared spectroscopy, which is tunable from $\lambda_{\text{ENZ}} = 1.25 \mu\text{m}$ to $\lambda_{\text{ENZ}} = 1.55 \mu\text{m}$ by thermal annealing after sputtering.”

Comment 3: “ITO is known to have graded refractive index as a function of film depth because ITO film commonly grows with a graded microstructure. Did the authors take this into account? If there is no grading, can the authors show that ellipsometry data modelling doesn’t require inclusion of refractive index grading to get a good fit?”

Our response: We thank the reviewer for this comment.

As noted by the reviewer, a gradient refractive index distribution can happen for thick ITO thin films. However, our ITO thin film samples have a thickness of only 50 nm . In addition, our sputtered ITO samples were post-treated by thermal annealing, which produced a more homogeneous distribution of refractive index in the vertical dimension.

In the **new** Figure S7, the fitting results further suggest no major difference in the permittivity with and without including a gradient refractive index distribution of ITO (panels a-b). The upper and lower layers of an ITO film show similar permittivity based on a graded ITO model (panel c). Using a homogeneous permittivity distribution for ITO thin films (thickness $\sim 50 \text{ nm}$), the ellipsometry fitting model captured the varying silica film thickness in the ITO/silica/ITO multilayers (panels d-f).

New Figure 17. The fitting of ITO permittivity. (a-c) A single ITO film. (d-f) The ITO/silica/ITO multilayer films fabricated by magnetic sputtering, where the silica thickness varies from 133.5 nm to 214.5 nm. Solid lines represent the measured signals and dashed lines show the ellipsometry fitting with a Drude model.

Change to manuscript: We added new results and discussions on ellipsometry fitting to the supporting information (new Figure S17).

Comment 4: “In Fig. 2b and its corresponding discussion (line 109), the authors say that the optical near field is enhanced within the ENZ layers. The following points require some clarification:

- (i) Typically, enhancement is in comparison with some reference which does not seem to be the case here. What is plotted in Fig. 2b seems to be the electric field intensities in the layers which doesn't necessarily mean enhancement.
- (ii) Even if one is to say that there is enhancement, the electric field in the inner ENZ layer is very small which contradicts the original statement that the optical near field is enhanced in both the ENZ layers.”

Our response: We thank the reviewer for this critical comment.

(1) In our FDTD modeling (Figure 2b), we set the initial electrical field amplitude of the pump source as one. Hence, the near field intensity with ENZ layers was plotted with respect to a unity input light intensity of the pump source ($|E|/|E_0|$). The optical field intensities within ENZ can be three orders of magnitude stronger compared to the input light source (Figure 2e).

(2) In Fig. 2b, the outer layer showed an enhancement factor of $|E|/|E_0| = 43.5$, and the inner layer showed an enhancement factor of $|E|/|E_0| = 18.2$. Note that the optical field intensity showed an oscillatory dependence on the interlayer separation distance (Figure 2c). For example, a stronger $|E|$ enhancement within the inner ENZ layer was shown in Figure 2c for a separation $l_0 = 450$ nm, and such a field intensity is comparable to that of the outer ENZ layer.

Change to manuscript: We revised the figure labeling and added related discussions to the manuscript (new Figure 2, page 6).

Comment 5: “In Fig. 2d, are the ENZ layers located on either side of the shaded region? This is not clear from the figure caption or the corresponding discussion and needs to be clarified.”

Our response: We thank the reviewer for suggesting more clarification on Figure 2d.

The shaded regions correspond to the two ENZ layers, where the left ENZ layer is sited on $x = 0.2 \mu\text{m}$ and the positioning of the right layer varies from $x = 0.4 \mu\text{m}$ to $x = 1.1 \mu\text{m}$.

Change to manuscript: We revised Figure 2d and added new discussions in the figure caption.

Comment 6: “Fig. 2d is not clear particularly in relation to Fig. 2c, where the fields in the individual ENZ layers are plotted as a function of spacer layer thickness. While l_a, l_b, l_c, l_d are clear in Fig. 2c, they are not clear in the way the phase data in Fig. 2d is presented. There are shaded regions in Fig. 2d which does not seem to correspond well with l_a, l_b, l_c, l_d (refer to point # 5). Perhaps there is a better way to represent this data.”

Our response: We thank the reviewer for this comment.

In Figure 2c, the x-axis shows the separation distance l between two ENZ layers. To be compared, the x-axis in Figure 2d represents the exact position of two ENZ layers in the FDTD modeling. We revised Figure 2d by recoloring the gray regions to be light blue to represent the ENZ layers. We also highlighted the distance l between two ENZ layers in Figure 2d.

New Figure 2. Comparison of near field oscillations between ENZ/silica/ENZ and air/silica/air multilayers.

Change to manuscript: We revised Figure 2 and added new discussions on page 6.

Comment 7: “How is the anti-correlated behavior in the electric field from Fig. 2 connected to the anti-correlated behavior in the spectral shift in Fig. 3b? A detailed physical explanation will be very useful.”

Our response: We thank the reviewer for encouraging us to elaborate more on the connections between Figure 2 and Figure 3b.

First, we would like to elaborate on the relationship between the near field enhancement and resonance intensity from the linear transmission spectra.

As transmission dips indicate the percentage of light extinction. Our modeling in Figures S1-2 suggests that the optical extinction comes partly from light reflection (35%) at the air/ENZ interface, and partly from light absorption (65%) within the ENZ thin film. The latter indicates trapped light within the ENZ thin film as manifested by the near-field enhancement, which is attributed to Berreman resonance.

The absorbed light by ENZ thin films is trapped within the films, and hence the light extinction ratio (**resonance intensity** change in Figure 3b) directly corresponds to the electric field intensity in the near field (**near field intensity** change in Figures S1c-d).

Figures S1. Origin of the transmission dip at the ENZ wavelength.

In addition, **new** modeling in Figure R1 further showed that resonance intensity in the transmission dip directly correlates with the intensity of electric near fields at resonance for a single-layer ENZ film. As the resonance intensity decreases with increased materials loss, the near-field enhancement within ENZ also decreases. Similarly, anti-correlated behavior in the electric field in Fig.2 agrees with the resonance intensity shift at ENZ wavelengths, as measured in Figure 3b (**new** Figure S12g-h).

Figure R1. Correlation between the resonance intensity in the transmission dip and the intensity of electric near-fields at ENZ resonance.

Change to manuscript: We added more discussions and new modeling results to the manuscript (**new** Figure S1, **new** Figure S12).

“The optical extinction at ENZ wavelength originates from both light reflection and absorption, the latter indicating trapped light within the ENZ thin film as manifested by the near-field enhancement, which is attributed to Berreman resonance (Supplementary Fig. S1c-f).”

Comment 8: “The simulated results in Supplementary Figures S12e and S12f have very poor agreement with the corresponding experimental observations of Fig. 3b. In particular, (i) the anti-correlated behavior is very minimal in the simulations, (ii) the spectral positions of the ENZ wavelengths are $\lambda_{\text{ENZ1}} = 1.4 \mu\text{m}$ and $\lambda_{\text{ENZ2}} = 1.9 \mu\text{m}$ experimentally while in the simulations, they are $\lambda_{\text{ENZ1}} = 1.3 \mu\text{m}$ and $\lambda_{\text{ENZ2}} = 1.5 \mu\text{m}$, and (iii) depth of modulation of the transmission spectrum at the ENZ points is very small in the experiment compared to the simulations. These are major discrepancies that need to be explained.”

Our response: We thank the reviewer for this critical comment. Here, we elaborate more on the correlation behavior in response to the comments accordingly.

(1) To better show the anti-correlated behavior of the resonance intensity at λ_{ENZ1} and λ_{ENZ2} , we replotted the transmission spectra in Figure S12f by calibrating the background transmission intensity at $1.15 \mu\text{m}$ ($l = 100 \text{ nm}$). As shown in the **new** Figure S12, the resonance intensity at λ_{ENZ1} , as quantified by the depth of the transmission dip, continuously decreased with an increased spacer separation from 100 nm to 300 nm. For λ_{ENZ2} , the resonance intensity continuously increased instead. Such evident anti-correlated behavior of resonant intensity indicated by the depth of transmission dip agrees well with the electric field intensity oscillation in Figure 2c.

New Figure S12. The anti-correlated behavior of the resonance intensity at λ_{ENZ1} and λ_{ENZ2} in the transmission spectra.

(2) Note that in our fabrication process of the ITO/silica/ITO multilayers, there is a limited tunability of the ENZ wavelength of the top ITO layer. Since thermal annealing was needed after sputtering to blueshift the ENZ wavelengths of fabricated ITO, a second high-temperature annealing of the ITO/silica/ITO multilayers would introduce wrinkling of the thin film samples.

In addition, considering the broad linewidths of ENZ resonances in experiments, separating the two ITO layers with two ENZ wavelengths helps differentiate the mode evolution from each other at different interlayer separations. Pumping at $1.4 \mu\text{m}$ and $1.7 \mu\text{m}$ still approaches the ENZ regions in ITO multilayers, as evidenced by the SHG measurement in Fig. 5a. The evolution of measured transmission spectra captured the first half of the oscillation period in Fig. 3c, which agrees with the modeled transmission spectra behavior in Figure S12.

(3) The measured depth of the transmission dips at ENZ wavelengths is smaller than that in the modeling due to the material loss of ITO thin films (new Figure S1). The imaginary part of the permittivity of ITO reduced the light trapping efficiency at the ENZ resonance.

New Figure S1. Influence of materials loss to the ENZ resonance lineshape and intensity.

Change to manuscript: We added new modeling results and discussions to the manuscript (new Figures S1, S12).

“In order to better show the anti-correlated behavior between the resonance intensity at λ_{ENZ1} and λ_{ENZ2} , the transmission spectra in Figure S12f were replotted by calibrating to the background transmission intensity at 1.15 μm ($l = 100 \text{ nm}$). The resonance intensity at λ_{ENZ1} , as quantified by the depth of the transmission dip, continuously decreased with an increased spacer separation from 100 nm to 300 nm. For λ_{ENZ2} , the resonance intensity continuously increased instead. Such evident anti-correlated behavior agrees well with the electric field intensity oscillation in Figure 2c. Note that the measured depth of the transmission dips at ENZ wavelengths is smaller than that in the modeling due to the material loss of ITO thin films.”

Comment 9: “Can the authors perform the measurement of the SHG intensity for wavelength 1.5 μm , at 450 degree and $d = 163 \text{ nm}$. This configuration will have ENZ resonance at 1.45 μm , quite close to the pump wavelength of 1.5 μm and will be a better confirmation of the room temperature result at $\lambda = 1.2 \mu\text{m}$ in Fig. 4c.”

Our response: We appreciate the reviewer for allowing us to discuss more on this topic.

First, for clarification on our nonlinear optical measurements, the single-layer graded ITO thin film was post-treated by thermal heating (300°C to 450°C) to blue-shift the ENZ wavelengths to the desired wavelengths, which produced a graded ENZ wavelength shift from 1.2 μm to 1.45 μm across the same sample at different thickness regions (Figure 1d). Afterwards, the SHG and transmission spectrum measurements were carried out at **room temperature**.

In Fig. 4c, we have measured the SHG intensity at both pump wavelengths $\lambda_{\text{pump}} = 1.2 \mu\text{m}$ (black curve) and $\lambda_{\text{pump}} = 1.5 \mu\text{m}$ (red curve) across the single-layer graded ITO thin film. The same ITO sample showed a thickness variation from 72 nm to 163 nm across the lateral dimension (new Figure S13), where the corresponding ENZ wavelength varied from 1.2 μm to 1.45 μm .

A major difference in SHG response was observed based on varying the pump wavelength λ . With $\lambda_{\text{pump}} = 1.2 \mu\text{m}$, the measured SHG signals showed a maximum intensity at $d = 72 \text{ nm}$. For $\lambda_{\text{pump}} = 1.5 \mu\text{m}$, the strongest SHG happens in the middle region instead.

New Figure S13. Scheme of the graded ITO thin film for SHG measurements in Figure 4.

Change to manuscript: For better clarification, we added a **new** scheme on the sample condition and the optical setup (**new** Figure S13). We added more discussions to the main text (page 10) and revised Fig. 4c by changing λ to λ_{pump} .

Comment 10: “Lines 347-350 (FTIR spectroscopy measurements) – if these measurements were done with an FTIR microscope, the angle of incidence is not a single value, rather a range of values determined by the NA of the microscope objective. In this case, how is the comparison with simulations made where the angle of incidence is taken to be a single value? For example, in Figs. S12e and S12f, the angle of incidence is mentioned to be 15 degrees while such a single value is not possible in the experiments performed with a microscope objective.”

Our response: We appreciate the reviewer for this comment.

Different from the microscope setup, our transmission spectra were taken within the FTIR integrating sphere chamber (Nocolet Continuum iS50 FT-IR microscope, Thermo Scientific).

The angle-resolved transmission was measured with a rotational sample holder in the beam path. As shown in the photo below, the samples were mounted on a rotational stage with a one-axis translation mount (Thorlabs, XF100). A **new** photo of the optical setup was added to the supporting information for the angle-resolved FTIR measurements.

Change to manuscript: We added the new photo of our angle-resolved FTIR measurement setup in the **new** Figure S11.

New Figure S11. Photo of the optical setup for measuring angle-resolved transmission spectra.

REVIEWER # 3:

Comment 1: “The manuscript entitled "Long-range Optical Coupling with Epsilon-near-zero Materials" by Wang et al. presents a compelling investigation into a multilayered structure (ENZ/Spacer/ENZ), where anti-correlated electromagnetic field intensities are explored through second harmonic generation (SHG) from wedged indium tin oxide (ITO) multilayers. While **the study addresses an interesting and timely topic**, I believe significant revisions are necessary before the work can be considered for publication. Below, I outline my detailed comments and suggestions that the authors should address in a revised version of the manuscript.”

Our response: We thank the reviewer for taking the time to review our manuscript and for appreciating the significance of our findings. We provided point-to-point responses to the reviewer’s comments as follows.

Comment 2: “Quantum Mechanical Analogy: The authors are encouraged to consider the following reference in their discussion of the quantum mechanical analogy, particularly when addressing resonant tunneling effects in ENZ structures: Caligiuri, Vincenzo, et al. "A semi-classical view on epsilon-near-zero resonant tunneling modes in metal/insulator/metal nanocavities." Nano Letters 19.5 (2019): 3151–3160. The novelty of the proposed approach should be clarified in light of the cited work.”

Our response: We thank the reviewer for suggesting additional references on this topic.

The difference between our work and the referred publication includes:

- (1) In the referred publication by Caligiuri *et al*, metal/insulator/metal was treated as a homogeneous, single ENZ layer with an effectively zero optical permittivity. The authors showed an analogy of such a system to quantum tunneling with a **single quantum barrier** (Figures 5e,f). To be compared, we studied the analogy of ENZ double layers to resonant quantum tunnelling in quantum **double barriers**, as shown in the scheme of Figure R3.
- (2) In addition, the **near-field distributions** are distinct between our double-layer ENZ system and the MIM system. The MIM structure sustains decaying fields within the metals with almost no field enhancement. In contrast, our double-layer ENZ system supports orders of stronger optical fields within the ENZ thin films, as driven by the continuity of the displacement D field at the materials interface.
- (3) Unique to our findings, **long-range coupling** exists between ENZ double layers up to hundreds of microns, together with anti-correlated optical near fields. Such interplay between near-field and far-field interactions in multilayer ENZ thin films, to our knowledge, has not been investigated in past studies.

Nano Letters, 19(5), 3151–3160, 2019
(single quantum barrier)

near-field distribution

Our work
(quantum double barriers)

near-field distribution

Figure R2. The mechanism difference between our work and the past literature on ENZ resonant tunneling.

Change to manuscript: We have added the suggested reference and related discussions to the main text (page 3), and highlighted the difference compared to our system.

“A metal/insulator/metal nanocavity acts effectively as a single ENZ layer, where the cavity resonance corresponds to ENZ eigenmodes that can be excited by resonant tunneling [new ref. 53]. Beyond a single layer, multiple layers of ENZ thin films offer a new degree of freedom to modulate optical coupling, while no work to our knowledge has investigated the interplay between near-field and far-field interactions”.

Comment 3: “Comparison with Recent Studies: The manuscript reports numerical and experimental studies on ENZ/spacer/ENZ configurations with varied spacer thicknesses. The authors should also discuss the following recent contribution and clarify how their work differs or advances the field:

Lio, Giuseppe Emanuele, et al. "Unlocking Optical Coupling Tunability in Epsilon-Near-Zero Metamaterials Through Liquid Crystal Nanocavities." *Advanced Optical Materials* 12.13 (2024): 2302483.”

Our response: We appreciate the reviewer for suggesting a new reference on this topic.

We have included this reference and discussions in the manuscript. Notably, the cited paper showed tuning optical coupling in two MIM layers with an effectively-zero permittivity, where the interlayer separation distance is controlled by a liquid crystal, and the optical interaction was indicated by the transmission spectra evolution. However, it differs from our system since no near-field intensity behavior or long-distance coupling properties were investigated.

Change to manuscript: We added the suggested reference and new discussions to the main text (new reference 58, page 3).

Comment 4: “Material Choice Justification: The rationale behind using an ITO/SiO₂/ITO structure instead of other alternatives, such as Ag/ITO/Ag, should be discussed. The latter may offer advantages in tuning the resonance without requiring annealing processes.”

Our response: We thank the reviewer for this comment.

The rationale for us to use an ITO/SiO₂/ITO structure is that ITO itself can serve as a single layer ENZ in the near-infrared (tunable between 1.2-1.8 μm), which is applicable for optical measurements with silicon based CCD, including transmission spectrum and second harmonic generation signals. To be compared, the ENZ wavelength of Ag is far into the UV regime (200-300 nm), which is not easily accessible for optical measurements.

Change to manuscript: We added related discussions to the main text.

Comment 5: “Field Confinement at Resonance: On page 4, the authors are requested to provide numerical simulations showing the electric field confinement within the SiO₂ layer at resonance and off-resonance conditions to support their interpretation.”

Our response: We thank the reviewer for this comment.

The strong field confinement and oscillation within the ENZ and dielectric layers were detailed in Figure S2b. As driven by the *D* field continuity at the materials interface under TM polarization, intense optical fields were observed within the ENZ layers in numerical simulations. In addition, we showed that the electric field oscillations with characteristic standing wave patterns exist in the SiO₂ spacer layer, which is sandwiched between two ENZ layers.

New Figure S7. Numerical simulations showing the electric field confinement within the ITO and SiO₂ layer at resonance (TM polarization) and off resonance (TE polarization).

Change to manuscript: We added new modeling results to the supporting information to show that the oscillatory standing waves exist in between the ENZ layers (new Figure S7).

Comment 6: “Clarification of Terminology: On page 5, lines 98–101, the expression “electric field displacement” is vague and should be clarified to avoid ambiguity.”

Our response: We thank the reviewer for encouraging us to elaborate more on the terminology.

Here, the electric displacement field is referred to as the D field in the Maxwell equation:

$$\nabla \cdot D = \rho_{e0}$$

which leads to the boundary condition for normal components:

$$D_{2\perp} - D_{1\perp} = \rho_{e0} = 0 \text{ (if no charge exists at the interface)}$$

$$\text{Hence, } \epsilon_0 E_{1\perp} = \epsilon_0 \epsilon E_{2\perp}$$

for ENZ materials where $\epsilon = 0$ and $E_{1\perp}$ is a nonzero number (at oblique angle incidence), the normal component of the E field $E_{2\perp}$ is strongly enhanced within the ENZ film.

Change to manuscript: We added more clarification to the terminology in the manuscript.

“For a single-layer ENZ thin film at oblique incidence, the continuity of the electric displacement field D at the ENZ/dielectric interface ($\nabla \cdot D = \rho_{e0}$) induces strongly enhanced optical fields inside the ENZ layer with characteristics of evanescent waves under TM polarization (Supplementary Movies 1-2).”

Comment 7: “Reference to Supplementary Material: On page 5, line 101, the transmittance spectrum mentioned—if referring to that shown in Supplementary Figure S1—should include a direct reference (e.g., “see Supplementary Fig. 1”).”

Our response: We thank the reviewer for this suggestion. We have revised the main text and included the reference figure (supplementary Fig. 1).

Change to manuscript: We revised the main text discussions on page 5.

“Based on finite-difference time-domain (FDTD) methods, an intense, sharp resonance at λ_{ENZ} appeared in the transmission spectrum (Supplementary Fig. S1a-b).”

Comment 8: “Resonance at Normal Incidence: The manuscript should explicitly explain why the proposed system does not exhibit any resonance at normal incidence. This is a fundamental aspect that warrants discussion in the main text.”

Our response: We thank the reviewer for this critical point.

Similar to our responses to comment 6, the continuity of the electric displacement field D is essential in producing the electric field enhancement.

The boundary condition for normal components leads to:

$$D_{2\perp} - D_{1\perp} = \rho_{e0} = 0 \text{ (if no charge exists at the interface)}$$

$$\text{Hence, } \epsilon_0 E_{1\perp} = \epsilon_0 \epsilon E_{2\perp}$$

for ENZ materials where $\epsilon = 0$ and $E_{1\perp}$ is a nonzero number (at oblique angle incidence), the normal component of the E field $E_{2\perp}$ is strongly enhanced within the film. In contrast, at normal incidence ($\theta_i = 0^\circ$), $E_{1\perp} = 0$, which also drives $E_{2\perp}$ to be zero. Hence, no field enhancement at the ENZ resonance exists.

We also confirmed that no ENZ resonance exists at normal incidence based on linear transmission spectra from FDTD simulations. Consistently, no SHG responses were observed in the nonlinear optics modeling (**new** Figure S16).

New Figure S16e. Comparison of transmission spectra and SHG intensity at normal incidence and oblique-angle incidence.

Change to manuscript: We added more modeling results and discussions to the manuscript (**new** Figure S16, **new** figure caption of Figure S2).

“The continuity of the electric displacement field D is essential in producing the electric field enhancement ($\nabla \cdot D = \rho_{e0}$, $D_{2\perp} - D_{1\perp} = \rho_{e0} = 0$, if no charge exists at the interface). Hence, $\epsilon_0 E_{1\perp} = \epsilon_0 \epsilon E_{2\perp}$. For ENZ materials, where $\epsilon = 0$ and $E_{1\perp}$ is a nonzero number (at oblique angle incidence), the normal component of the E field $E_{2\perp}$ is strongly enhanced within the film. In contrast, at normal incidence ($\theta_i = 0^\circ$), $E_{1\perp} = 0$, which also drives $E_{2\perp}$ to be zero and hence, no strong field enhancement exists.”

Comment 9: “Clarity of Figures: Figures 2b and 2e lack clarity in depicting the structure. The boundaries of the ENZ layers, the spacer, and the surrounding medium (e.g., air) are not clearly delineated. A clearer schematic is recommended.”

Our response: We appreciate the reviewer for this suggestion.

We have marked the boundaries of the ENZ layers, the spacer, and the surrounding medium (the same as in panel a) in the new Figure 2.

New Figure 2. Near-field distribution of optical coupling between two ENZ layers.

Change to manuscript: We included the new Figure 2 in the main text.

Comment 10: “Electric Field Maps: To enhance understanding of light confinement, the authors are encouraged to include electric field distribution maps corresponding to the configuration in Figure 3, particularly for selected incident angles at both λ_{ENZ1} and λ_{ENZ2} .”

Our response: We thank the reviewer for this comment.

We have added the new electric field distribution maps corresponding to Figure 3. Notably, the two ITO layers showed either constructive or destructive interference as a function of the interlayer separation l .

Figure R3. Electric field distribution ($|E|/|E_0|$) in ENZ multilayers at λ_{ENZ1} and λ_{ENZ2} at the incident angle $\theta_i = 15^\circ$ and interlayer separation $l = 290$ nm.

Change to manuscript: We added new electric field distribution maps corresponding to Figure 3 in the supporting information.

Comment 11: “Stylistic Suggestion: The sentence spanning lines 172–174 on page 9 appears redundant and may be merged with the preceding statement for improved readability and conciseness.”

Our response: We thank the reviewer for this suggestion. We have revised the redundant statements on page 9.

Change to manuscript: We revised the main text as follows (page 10):

“Wavelength-dependent SHG measurement showed a maximum SHG intensity when the pump wavelength was close to λ_{ENZ} (Figure S15).”

Comment 12: “SHG Response Interpretation: In Figure 4b, the SHG peaks appear unshifted even when the cavity thickness is altered, despite pumping at λ_{ENZ1} . This seems inconsistent with the observations in Figure 5a, where SHG peaks shift. While a reduction in intensity is acknowledged, a wavelength shift might also be expected. This discrepancy should be discussed in greater detail.”

Our response: We thank the reviewer for this comment.

For clarification, Figure 4b shows the measured SHG results with a single, fixed pump wavelength at λ_{pump} (1.2 μm or 1.5 μm). Since the pump wavelength remains fixed, the generated SHG signals happen at doubled frequencies with no spectral shift. In the meantime, the altered ITO thickness produced a shifted ENZ wavelength at different sample regions, hence stronger SHG intensity happens when the pump wavelength is in resonant with ENZ mode and when the film thickness is smaller ($d = 72$ nm for a pump at $\lambda_{pump} = 1.2$ μm).

In contrast, Figure 5a represents the case of pumping a single ITO film with varying pump wavelengths (from 1.2 μm to 1.44 μm), which suggests that the maximum SHG happens when the pump wavelength is resonant with the ENZ wavelength. We have included new modeling results to show varying SHG wavelengths with varying pump wavelengths for a single ITO layer (new Figure S16).

New Figure S16. Modeling of the shifted SHG peaks from ENZ thin film with a varying pump wavelength.

Change to manuscript: We added related discussions and new modeling results to the supplementary information (new Figure S16).

“Varying pump wavelengths induced varying SHG wavelengths, and no SHG signals exist with a pump at normal incidence (Supplementary Figs. 16d-e).”

Comment 13: “Conclusion Revision: The conclusions section presently overstates the applicability of the proposed structure to quantum applications. A more balanced summary focusing on the actual findings would improve the overall credibility and focus of the manuscript.”

Our response: We thank the reviewer for this comment. We have revised as below to emphasize more on the major findings in the conclusion paragraph.

Change to manuscript: We have revised the discussions in the conclusion paragraph and highlighted new possibilities offered by photonic crystals for long-range optical coupling.

“Inspired by these findings, we anticipate that ENZ multilayers may serve as an optical ruler with a deep subwavelength spatial resolution to sense optical and biomedical environments. Replacing ITO materials with rationally designed photonic crystals [*Nature Materials* 10, 582–586 (2011)] allows for achieving effectively zero permittivity with lower material loss, which opens new prospects in nonlinear optics.”

Comment 14: “In Methods section, particularly concerning SHG, the authors should specify the pulsed laser power used and the corresponding beam size, to ensure reproducibility and proper comparison with other studies.”

Our response: We thank the reviewer for encouraging us to elaborate more on the optical setup.

We have included in the Method section that the pump beam spot was focused by an optical lens (focal distance $f = 3$ cm) on the ITO samples with a 20- μm diameter spot. For Figure 5, the measured SHG signals at $\lambda = 1.4$ μm were obtained with a pump power density of 541 W/cm^2 . For $\lambda = 1.7$ μm , the pump power density was 73 W/cm^2 . The pump power density in total varied from 64 W/cm^2 to 636 W/cm^2 .

Change to manuscript: We added the information on the pump pulse power and energy in the Method section (page 18).

“The laser beam was focused by an optical lens (focal distance $f = 3$ cm) on the ITO samples with a 20- μm diameter circular spot at an incident angle of 45° . The average pump power density for SHG measurements varied from 64 W/cm^2 to 636 W/cm^2 .”

REVIEWER # 4:

Comment 1: “This paper investigates long-range optical interactions in epsilon-near-zero (ENZ) materials, drawing a compelling analogy to quantum tunneling in electronic systems. The **most significant findings** can be summarized as follows:

(a) The electric field intensities within ENZ layers exhibit anti-correlated oscillations as the interlayer separation varies;

(b) This behavior is probed through second harmonic generation (SHG), a nonlinear optical process in which two photons combine to emit a single photon at twice the original frequency.

These results are both relevant and well-supported by robust experimental data and numerical simulations. In my opinion, **the manuscript is highly appropriate for the audience of *Nature Communications***, and I recommend acceptance after addressing the following minor point.”

Our response: We thank the reviewer for taking the time to review our manuscript and for appreciating the significance of our findings.

Comment 2: “To interpret the observed optical behavior, you propose an analogy with resonant quantum tunneling through a double-barrier system. While this comparison is insightful, I believe it is not entirely novel, as a similar perspective was already presented in the following reference:

Caligiuri, V., Palei, M., Biffi, G., Artyukhin, S., & Krahne, R. (2019). A semi-classical view on epsilon-near-zero resonant tunneling modes in metal/insulator/metal nanocavities. *Nano Letters*, 19(5), 3151–3160.

It would strengthen the manuscript to more clearly highlight both the similarities and the novel aspects of your approach in relation to this previous work. If relevant, clarifying how your results expand on or differ from the viewpoint proposed by Caligiuri et al. could further enhance the contribution of your paper.”

Our response: We appreciate the reviewer for suggesting related references on this topic.

The difference between our work and the referred publication includes:

(1) In the referred publication by Caligiuri *et al*, metal/insulator/metal was treated as a homogeneous, single ENZ layer with an effectively zero optical permittivity. The authors showed an analogy of such a system to quantum tunneling with a **single quantum barrier** (Figures 5e,f). To be compared, we studied the analogy of ENZ double layers to resonant quantum tunnelling in quantum **double barriers**, as shown in the scheme of Figure R2.

(2) In addition, the **near-field distributions** are distinct between our double-layer ENZ system and the MIM system. The MIM structure sustains decaying fields within the metals with almost no field enhancement. In contrast, our double-layer ENZ system supports orders of stronger optical fields within the ENZ thin films, as driven by the continuity of the displacement D field at the materials interface.

(3) Unique to our findings, **long-range coupling** exists between ENZ double layers up to hundreds of microns, together with anti-correlated optical near fields. Such interplay between near-field and far-field interactions in multilayer ENZ thin films, to our knowledge, has not been investigated in past studies.

near-field distribution

near-field distribution

Figure R2. The mechanism difference between our work and the past literature on ENZ resonant tunneling.

Change to manuscript: We have added the suggested reference and related discussions to the main text (page 3), and highlighted the difference compared to our system.

“A metal/insulator/metal nanocavity acts effectively as a single ENZ layer, where the cavity resonance corresponds to ENZ eigenmodes that can be excited by resonant tunneling [new ref. 53]. Beyond a single layer, multiple layers of ENZ thin films offer a new degree of freedom to modulate optical coupling, while no work to our knowledge has investigated the interplay between near-field and far-field interactions”.

REVIEWER # 2:

Comment 1: “The In this manuscript, Wang et al., show the analogue of long-range resonant electron quantum tunnelling in optical systems composed of epsilon-near-zero (ENZ) thin films. The authors have given a point-by-point response to my previous review comments.

While I am satisfied with the response provided by the authors to most of my previous queries, there is a concern regarding one of the aspects. In response to point#3 of comment#8, the authors had written that the transmission dips at ENZ wavelengths in experimental data is smaller than in the simulation due to the material loss of the ITO thin film. While I agree with this, can the authors not perform the simulation by taking into account the material loss? They do have the experimentally measured the optical constants of ITO film (Figure S17). Since the modulation of transmission dips is a crucial piece of data that supports the assertion of this work, I feel the authors should redo the simulation with the measured optical constants of ITO which I would expect should give a smaller depth of modulation consistent with the experimental data.”

Our response: We thank the reviewer for this comment.

The measured depth of the transmission dips at ENZ wavelengths is smaller than that in the modeling due to the material loss of ITO thin films. The imaginary part of the permittivity of ITO led to reduced light trapping efficiency at ENZ resonance. Here, we redid the simulation of transmission spectra of a single ENZ thin film with its optical permittivity ($\epsilon_i = 0.1$) closer to the measured value of ITO. The result showed a smaller depth of transmission intensity modulation, which is consistent with the experimental data.

Change to manuscript: We added the new modeling result and related discussions to the manuscript (new Supplementary Fig. 1).

New Supplementary Fig. 1. Influence of materials loss to the ENZ resonance lineshape and intensity.

In this manuscript, Wang et al., show the analogue of long-range resonant electron quantum tunnelling in optical systems composed of epsilon-near-zero (ENZ) thin films. While the proposition is interesting, in my view, the results do not fully attest to this and therefore, the manuscript is not yet at the level for publication in Nature Communications. There are several major aspects that need to be addressed as listed below:

1. There is a mismatch between data in Figs. 1c and 1d. In Fig. 1c, the ENZ point is at $\lambda = 1.2 \mu\text{m}$ whereas in Fig. 1d, the dip in transmission that corresponds to the ENZ point occurs at $\lambda = 1.9 \mu\text{m}$ (line 92 of manuscript text). This needs to be clarified.
2. ITO is known to have graded refractive index as a function of film depth because ITO film commonly grows with a graded microstructure. Did the authors take this into account? If there is no grading, can the authors show that ellipsometry data modelling doesn't require inclusion of refractive index grading to get a good fit?
3. In Fig. 2b and its corresponding discussion (line 109), the authors say that the optical near field is enhanced within the ENZ layers. The following points require some clarification:
 - (i) Typically, enhancement is in comparison with some reference which does not seem to be the case here. What is plotted in Fig. 2b seems to be the electric field intensities in the layers which doesn't necessarily mean enhancement.
 - (ii) Even if one is to say that there is enhancement, the electric field in the inner ENZ layer is very small which contradicts the original statement that the optical near field is enhanced in both the ENZ layers.
4. In Fig. 2d, are the ENZ layers located on either side of the shaded region? This is not clear from the figure caption or the corresponding discussion and needs to be clarified.
5. Fig. 2d is not clear particularly in relation to Fig. 2c, where the fields in the individual ENZ layers are plotted as a function of spacer layer thickness. While l_a, l_b, l_c, l_d are clear in Fig. 2c, they are not clear in the way the phase data in Fig. 2d is presented. There are shaded regions in Fig. 2d which does not seem to correspond well with l_a, l_b, l_c, l_d (refer to point # 4). Perhaps there is a better way to represent this data.
6. How is the anti-correlated behaviour in the electric field from Fig. 2 connected to the anti-correlated behaviour in the spectral shift in Fig. 3b? A detailed physical explanation will be very useful.
7. The simulated results in Supplementary Figures S12 e and S12f have very poor agreement with the corresponding experimental observations of Fig. 3b. In particular, (i) the anti-correlated behaviour is very minimal in the simulations, (ii) the spectral positions of the ENZ wavelengths are $\lambda_1 = 1.4 \mu\text{m}$ and $\lambda_1 = 1.9 \mu\text{m}$ experimentally while in the simulations, they are $\lambda_1 = 1.3 \mu\text{m}$ and $\lambda_1 = 1.5 \mu\text{m}$, and (iii) depth of modulation of the transmission spectrum at the ENZ points is very small in the experiment compared to the simulations. These are major discrepancies that need to be explained.

8. Can the authors perform the measurement of the SHG intensity for wavelength 1.5 μm , at 450°C and $d = 163 \text{ nm}$. This configuration will have ENZ resonance at 1.45 μm , quite close to the pump wavelength of 1.5 μm and will be a better confirmation of the room temperature result at $\lambda = 1.2 \mu\text{m}$ in Fig. 4c.

9. Lines 347-350 (FTIR spectroscopy measurements) – if these measurements were done with an FTIR microscope, the angle of incidence is not a single value, rather a range of values determined by the NA of the microscope objective. In this case, how is the comparison with simulations made where the angle of incidence is taken to be a single value? For example, in Figs. S12e and S12f, the angle of incidence is mentioned to be 15° while such a single value is not possible in the experiments performed with a microscope objective.